# Baseline Susceptibility and Laboratory Selection of Resistance to Bt Cry1Ab Protein of Chinese Populations of Yellow Peach Moth, *Conogethes punctiferalis* (Guenée)

**DOI:** 10.3390/toxins13050335

**Published:** 2021-05-06

**Authors:** Su Mon Shwe, Sivaprasath Prabu, Yu Chen, Qincheng Li, Dapeng Jing, Shuxiong Bai, Kanglai He, Zhenying Wang

**Affiliations:** State Key Laboratory for Biology of Plant Diseases and Insect Pests, Institute of Plant Protection, Chinese Academy of Agricultural Sciences, No. 2, West Yuanmingyuan Road, Beijing 100193, China; sumoshwe.ento@gmail.com (S.M.S.); sivaprasathibt@gmail.com (S.P.); ychen_007@126.com (Y.C.); ashofmournhold@163.com (Q.L.); jingfly6@163.com (D.J.); sxbai@ippcaas.cn (S.B.); klhe@ippcaas.cn (K.H.)

**Keywords:** baseline susceptibility, *Conogethes punctiferalis*, Cry1Ab, diagnostic concentration, resistance monitoring

## Abstract

Yellow Peach Moth (YPM), *Conogethes punctiferalis* (Guenée), is one of the most destructive maize pests in the Huang-Huai-Hai summer maize region of China. Transgenic Bacillus thuringiensis (Bt) maize provides an effective means to control this insect pest in field trials. However, the establishment of Bt resistance to target pests is endangering the continued success of Bt crops. To use Bt maize against YPM, the baseline susceptibility of the local populations in the targeted areas needs to be verified. Diet-overlay bioassay results showed that all the fourteen YPM populations in China are highly susceptible to Cry1Ab. The LC_50_ values ranged from 0.35 to 2.38 ng/cm^2^ over the two years of the collection, and the difference between the most susceptible and most tolerant populations was sevenfold. The upper limit of the LC_99_ estimates of six pooled populations produced >99% larval mortality for representative eight populations collected in 2020 and was designated as diagnostic concentrations for monitoring susceptibility in YPM populations in China. Hence, we evaluated the laboratory selection of resistance in YPM to Cry1Ab using the diet-overlay bioassay method. Although the resistant ratio was generally low, YPM potentially could evolve resistance to Cry1Ab. The potential developmentof resistance by target pests points out the necessity to implement resistance management strategies for delaying the establishment of pest resistance to Bt crops.

## 1. Introduction

*Bacillus thuringiensis* (Berliner), a ubiquitous Gram-positive and endospore-forming soil bacterium, produce insecticidal crystal proteins during the sporulation phase of its growth cycle [1,2]. It has been using as an alternative to chemical insecticides to control the critical agricultural pests in the orders Lepidoptera, Coleoptera, and Diptera [3,4,5]. Since 1996, transgenic maize expressing Bt insecticidal proteins has been commercialized and used successfully to manage corn borer problems. It is widely grown in the key maize growing regions such as the USA, Brazil, Argentina, Canada, South Africa, and the Philippines and the estimated acreage reached 58.9 million hectares in 2018 [6].

Planting of Bt-transgenic maize can suppress target pest populations, reduce chemical insecticide use, and increase economic benefits to growers [7,8,9,10]. Although the environmental and economic benefits of planting Bt transgenic maize are widely recognized, concernshave been raised regarding possible insectsresistance to Bt crops [11]. Laboratory and field selection studies have presented that the potential to develop Bt toxin resistance is widespread among insect pest species [12,13,14,15,16,17,18,19]. Consequently, the growing of Bt crops must be carefully managed to delay insect resistance and prolong Bt productslifetime.

To maintain the effectiveness of Bt maize, the implementation of robust and experiment-based insect resistance management strategies is essential for target pest management [20]. These management strategies should include the effective resistance monitoring program, which can perceive as the evolution of resistance at an initial stage that will allow timely information to develop appropriate management decisions [21]. Before the commercial adoption of Bt maize, the development of target pests’ baseline susceptibility to Bt protein is crucial for establishing effective resistance management and monitoring program [22]. These data establish a diagnostic concentration used to control 99% of susceptible populations for resistance monitoring. [23,24]. Therefore, the implementation of resistance monitoring strategies would provide early warning information to breeders and growers.

There are six major maize-growing regions in China; North Spring Corn Region, Huang-Huai-Hai Summer Maize Region, Southeast Hilly Maize Region, Southwest Hilly Maize Region, Northwest Inland Maize Region, and Qing-Zang Plateau Maize Region. The Huang-Huai-Hai Summer Maize Region is the second most extensive region and estimates for up to 37% of China’s cumulative maize production acreage [25,26]. Among the pests damaging to maize, the Asian corn borer, *Ostrinia furnacalis* (Guenée) (Lepidoptera: Crambidae), yellow peach moth, *Conogethes punctiferalis* (Guenée) (Lepidoptera: Crambidae), fall armyworm, *Spodoptera frugiperda* (J.E. Smith) (Lepidoptera: Noctuidae), cotton bollworm, *Helicoverpa armigera* (Hübner) (Lepidoptera: Noctuidae), black cutworm, *Agrotis ipsilon* (Hufnagel) (Lepidoptera: Noctuidae), and oriental armyworm, *Mythimna separata* (Walker) (Lepidoptera: Noctuidae), count as the significant Lepidopteran pests from seedling stage to mature of maize [27,28,29,30].

Yellow peach moth (YPM) is a polyphagous pest attacking several crops, including peach, chestnut, sunflower, sorghum, and maize [31,32,33], and broadly spread in South and East Asia, Australia, and Papua New Guinea [34]. In China, it distributed from north to south across the country, including Liaoning, Shaanxi, Shanxi, Hebei, Beijing, Tianjin, Henan, Shandong, Anhui, Jiangsu, Jiangxi, Zhejiang, Fujian, Taiwan, Guangdong, Hainan, Guangxi, Hunan, Hubei, Sichuan, Yunnan, and Tibet [35]. In recent years, YPM has become the most destructive maize pest especially in the Huang-Huai-Hai summer maize region in China [36]. It mainly feeds on the maize ear, inducing ear rot, reducing grain quality, and causing severe economic yield losses [37]. The larva of YPM is an atypical generalist and potentially can spread to long distances, and its multivoltine nature helps the YPM maintain its population constantly [38]. Although chemical pesticides are currently the significant control measure of managing YPM, there are extreme consequences because of their adverse effects on non-target pests, beneficial organisms, human health, and the environment [39]. Therefore, using biological insecticides that are the product of Bt is an alternative to chemical insecticides to control the critical agricultural pests consisting of orders Lepidoptera, Coleoptera, and Diptera [3,4,5]. Accordingly, before Bt maize’s comprehensive commercial cultivation in China, it plays a crucialrole in formulating new strategies to control this pest and establish a science-basedpest management program. Therefore, this study was taken to establish YPM’s baseline sensitivity to Cry1Ab in distinct geographical populations in China. Here we describe the first survey of baseline susceptibility to Cry1Ab of different YPM geographical populations across major maize growing regions in China, which will be an initial step for the future resistance monitoring program.

## 2. Results

### 2.1. Baseline Susceptibility of YPM to Bt Cry1Ab Protein

Fourteen YPM larvae populations were collected in the non-Bt maize fields from China’s different geographical areas in 2019 and 2020 (Table 1). The largest populations were collected in the Shandong, Henan, Hebei, and Anhui provinces from the Huang-Huai-Hai summer maize production region which is the China’s second largest maize-growing region (Figure 1). The susceptibility of YPM larvae to Cry1Ab for fourteen field and lab populations was presented in Table 2. Diet-overlay bioassay results showed that all the fourteen YPM populations in China are highly susceptible to Cry1Ab. The LC_50_ values ranged from 0.35 to 2.38 ng/cm^2^ for the populations over the two years of the collection, while the LC_99_ values ranged between 40.13 and 409.65 ng/cm^2^. The population from Luohe had the lowest LC_50_ value (0.35 ng/cm^2^); it was not significantly different from Luoyang (0.55 ng/cm^2^) and Jiyuan population (0.65 ng/cm^2^), all from Henan province. Tangshan population produced the highest LC_50_ value (2.38 ng/cm^2^), followed by Handan (2.19 ng/cm^2^), Suzhou (2.14 ng/cm^2^), and Linyi (1.72 ng/cm^2^) populations.

### 2.2. YPM Larval Growth Inhibition to Bt Cry1Ab Protein

For the larval growth inhibition, YPM larvae showed high susceptibility to Cry1Ab in all tested populations (Table 3). The larval growth inhibition recorded >50% in the Cry1Ab lowest concentration (0.004 ng/cm^2^) and above 99% larval growth inhibition was observed in the concentration of 13.68 ng/cm^2^ and 68.42 ng/cm^2^, respectively.

### 2.3. Development of Diagnostic Concentration

After pooling the six populations collected in 2019 in the F2 generation, the LC_99_ for Cry1Ab was 43.33 ng/cm^2^, with a lower and upper limit 24.28 ng/cm^2^ and 92.73 ng/cm^2^, respectively. The upper limit of 92.73 ng/cm^2^ exhibited larval mortality >99% for all eight populations collected in 2020 (Table 4). However, both LC_99_ (24.28 ng/cm^2^) and the lower limit of 43.33 ng/cm^2^ produced larval mortality <99% for eight populations. Based on this, the upper limit of the LC_99_ estimates could be potentially suitable as the diagnostic concentration for future monitoring work of Cry1Ab resistance in the YPM population.

### 2.4. Laboratory Selection

Large-scale diet-overlay bioassays similar to those described above were conducted for the selection experiments. As mentioned in the methodology, the laboratory strain was steadily selected with increasing doses of Cry1Ab from 1.08 to 6.16 ng/cm^2^ for the 14th generation (Table 5). Resistance ratio increased 2.26-fold in 3rd generation, 3.30-fold in 6th generation, 4.76-fold in 9th generation, 5.42-fold in 11th generation, 6.08-fold in 13th generation, and after the 14th generation of selection, the resistance ratio increased by 7.10-fold, respectively.

## 3. Discussion

Before the broad commercial deployment of Bt maize, the establishment of baseline susceptibility of target pests to Bt protein is the first step for generating an effective resistance monitoring and management program [22]. In the present study, we conducted the diet overlay bioassays to verify the baseline susceptibility of YPM to Cry1Ab from fourteen diverse geographical populations of China. In general, Cry1Ab proteins have observed high toxicity against the species of genus *Ostrinia* [25,41,42,43]. A previous report documents that YPM is highly susceptible to Cry1Ab proteins than other tested proteins [44]. Zhang et al., 2010 [45] concluded that high genetic similarity among 11 populations but Shandong populations showed lower genetic diversity than others and biggest genetic distance was identified between Sichuan and Shandong populations. Here the LC_50_ of Shandong populations were in general higher than those from inner land provinces (although not significant).

The natural variation in baseline susceptibility associate with various factors such as population vigor, the source of the Bt proteins, bioassay method, environmental condition, time of exposure, and artificial diet [46,47,48,49]. Several studies have shown minor to moderate variation in the Crambidae family to Cry1Ab among various countries. The Cry1Ab susceptibility varied less than 10-fold among ten different *O. furnacalis* populations across China [25]. Alcantara et al. (2011) reported less than sixfold limited variability in Cry1Ab susceptibility of *O. furnacalis* populations in the Philippines. Another study in Vietnam also showed threefold variations in the LC_50_ estimate values of *O. furnacalis* to Cry1Ab [42]. Li et al. (2020) reported that no significant natural variation in susceptibility to Cry1Ab was found among 15 field populations of *O. furnacalis* in China. Moreover, Marçon et al. (1999) did not find a relatively high level of variability (<4 fold) in susceptibility to Cry1Ab for *O. nubilalis*. Furthermore, the two populations of *O. nubilalis* in Spain exhibited a relatively low level of variability in susceptibility to the Cry1Ab [50]. However, the ranges of LC_50_ among populations in response to Cry1Ac protein were 16-fold for *Helicoverpa zea* in the USA [51]. *H. armigera* population from China displayed a 100-fold high level of natural variability in response to Cry1Ac protein [52]. In this study, the LC_50_ values among fourteen different populations ranged from 0.35 to 2.38 ng/cm^2^ with 7-fold variations, while the LC_99_ values ranged between 40.13 and 409.65 ng/cm^2^ with 10-fold variations. These findings collectively indicated that the Crambidae family exhibits relatively little variability in response to the Cry1A protein compare with those of the Noctuidae.

The high sensitivity growth inhibition may be considered as an additional indicator for diagnostic concentration response in future monitoring work [53]. In our study, severe larval growth inhibition of YPM larvae byCry1Ab was detected in the different geographical populations (Table 3). The results showed that >50% larval growth inhibition in the Cry1Ab lowest concentration (0.004 ng/cm^2^) and >99% larval growth inhibition in the concentration of 13.68 ng/cm^2^ and 68.42 ng/cm^2^, respectively. Accordingly, when increasing the Cry1Ab concentration, YPM larval weight also reduced after one week of treatment. Marçon et al. (1999) documented that high larval stunting was showed in the low concentration of Cry1Ab is common among tested species, including *Ostrinia* spp. Furthermore, in Vietnam, the *Ostrinia* population exhibited >90% larval stunting in a concentration of above 0.82 ng/cm^2^ [42]. In addition, high larval growth inhibition was observed to Bt Cry1A protein in the noctuid in different countries [52,54,55,56,57].

To monitor insect resistance’s evolution to Bt proteins, the diagnostic dose’s development and estimation are necessary steps in the insect resistance monitoring program. In some countries, the LC_99_ values have been applied to develop a diagnostic dose for monitoring the evolution of insect resistance to Cry1 proteins [23,53,58]. In this study, we collected six populations in 2019 to designate the diagnostic concentration of YPM to Cry1Ab. The LC_99_ estimates for Cry1Ab were 43.33 ng/cm^2^, with a lower and upper limit of 24.28 ng/cm^2^ and 92.73 ng/cm^2^, respectively. The upper limit of the LC_99_ (92.73 ng/cm^2^) was selected as the candidate diagnostic concentration because of exhibited consistent larval mortality above 99% when treated on eight representative YPM populations in China (Table 4). A study from 2020, the LC_99_ estimate of Cry1Ab was 93 ng/cm^2^, killed >99% of *O. furnacalis* populations, and was identified as the diagnostic dose for monitoring susceptibility of *O. furnacalis* in China [59]. These observations suggested that the development of effective diagnostic concentration could be successfully applied to control both YPM and *O. furnacalis* in future monitoring work.

The establishment of Bt resistance to target pests is threatening to the continuous success of Bt crops. Several studies found *O. furnacalis* population has been evolved resistant to Bt Cry1 protein after several generations in the laboratory selection [60,61,62]. Besides, both a laboratory and field study observed that some noctuid could evolve resistance to Cry1Ab [63,64]. We attempted the laboratory selection of resistance in YPM to Cry1Ab using the diet-overlay bioassay method. Resistance ratio increased 2.26-fold in 3rd generation, 3.30-fold in 6th generation, 4.76-fold in 9th generation, 5.42-fold in 11th generation, 6.08-fold in 13th generation, and after the 14th generation of selection, the resistance ratio increased by 7.10-fold, respectively (Table 5). Although the resistant proportion was generally low, YPM potentially could evolve resistance to Cry1Ab. The possible evolution of resistance to target pests points out the necessity to implement resistance management strategies for delaying the establishment of pest resistance to Bt crops.

## 4. Conclusions

We report here on the baseline susceptibility of distinct geographical populations of YPM to Cry1Ab, which is crucial to establish an effective resistance monitoring program for YPM in China. Our findings collectively pointed that the Crambidae family exhibits relatively little variability in response to the Cry1A protein compare with those of the Noctuidae. Additionally, the upper limit of the LC_99_ (92.73 ng/cm^2^) was verified as the candidate diagnostic concentration. Presently, a resistant strain of YPM is being developed in the laboratory and can be potentially used for testing the efficacy of the diagnostic technique. The diagnostic concentration validation study should compare the YPM resistant population and the field-collected population to future monitoring programs.

## 5. Materials and Methods

### 5.1. Insects Collection

Fourteen YPM field populations were collected from the maize fields in seven provinces, including Shandong, Hebei, Henan, Shaanxi, Beijing, and Anhui provinces across the Huang-Huai-Hai summer maize region and Sichuan Province from Southeast Hilly Maize Region in China. Approximately 200–350 larvae (4th–5th instar) were collected from maize ears at each location in autumn before maize harvest. Only the Chengdu population was collected about 65 larvae because of less population in that collected area. Collection sites were chosen based upon their infestation history and were made in 2019 and 2020 (Figure 1).

### 5.2. Laboratory Rearing of Field-Collected YPM 

Field collected larvae were reared on the freshly prepared artificial diet until pupation [65]. The pupae were transferred to mating cages for adult emergence. Inside the adult rearing cages, the apple substrate wrapped with gauze was placed for oviposition. A cotton swab soaked in 15% honey solution was provided for the adults and renewed daily. The YPM was maintained in the rearing room at 27 ± 1 °C, 60–70% relative humidity (RH), and a 16:8 h (L:D) photoperiod. Newly hatched neonate larvae (<24 h) from the F1 generation were used for diet-overlay bioassays and diagnostic concentration assays. The Jiyuan and Chengdu populations were raised to F2 and F3 generation as not enough for further experiment. YPM laboratory strain was collected in a maize field where Bt spraying is not practiced at Langfang, Hebei Province, China. The collected larvae were reared using the same method mentioned above for eighteen generations without exposure to Bt protein.

### 5.3. Baseline Susceptibility Bioassay 

Trypsin-activated Cry1Ab protein was purchased from Envirologix (Portland, OR, USA). Cry1Ab protein’s insecticidal activity against YPM larvae was assessed in concentration-response bioassays by applying Cry1Ab preparations on the artificial diet’s surface. Seven protein concentrations ranged from 0.004 to 68.42 ng/cm^2^ (protein/diet). All bioassays were performed by transferring neonate larvae (<24 h) with a fine brush to contaminated diet surfaces in a 24-well plate. The plates were sealed with a membrane (Cat# 3M-9733, Minnesota Mining, and Manufacturing Company, Saint Paul, MN, USA) perforated with a sharp pin on each cell to provide aeration. Plates were maintained in the rearing room at 27 ± 1 °C, 60–70% RH, and a 16:8 h (L:D) photoperiod. Three replicates of 24 larvae per concentration (*n* = 72) were applied for each Bt protein dose, and double-distilled water (no Bt) was applied for the control. The larval mortality and larval weight were scored one week after treatment. The larvae that failed to move when prodded with a fine pin were considered dead.

### 5.4. Establishment of Diagnostic Concentrations of Bt Cry1Ab Protein

Diagnostic concentrations were established with the pooled population from six locations during 2019 field collections. A pooled analysis using the logistic option in the PROBIT procedure was performed across populations to estimate LC_99_ and its lower and upper limit, which was considered the diagnostic concentration for future resistance monitoring purposes. To evaluate the LC_99_, lower limit, and the upper limit for use as the diagnostic concentration, eight YPM populations were collected in Shandong, Hebei, Henan, Beijing, Anhui, and Sichuan provinces in 2020. About 240 neonate larvae from each population were tested with each of the three candidates’ diagnostic concentrations of Cry1Ab using the diet overlay bioassay described above.

### 5.5. Laboratory Selection

Laboratory strains of YPM originated from the maize field at Langfang experimental station of Chinese Academy of Agricultural Sciences, Hebei Province, China, and reared on the artificial diet in the laboratory for eighteen generations without exposure to Bt proteins. Diet overlay bioassay was conducted to evaluate this strain’s susceptibility to Cry1Ab before conducting a selection experiment, and the LC_50_ estimate was 1.08 ng/cm^2^ (toxin/diet). The population was initially treated with LC_50_ value of Cry1Ab for ten days and transferred to an untreated artificial diet to complete its life cycle. It was selected steadily with increasing the concentrations of Cry1Ab (2.05 ng/cm^2^) for 6th generation, (3.12 ng/cm^2^) for 9th generation, and (4.32 ng/cm^2^) for 11th generation and then maintained at (6.16 ng/cm^2^) for 14th generation, respectively. The resistance ratio was calculated LC_50_ value of the Cry1Ab resistance strain divided by LC_50_ value of the Cry1Ab susceptible strain.

### 5.6. Statistical Analysis

Fifty percent lethal concentration (LC_50_) with 95% fiducial limits (FL) and LC_99_ were determined using the PoloPlus (v 1.0, LeOra Software, Parma, MO, USA) for Cry1Ab protein and the slope for all bioassays by probit analysis. The difference between LC_50_ values and LC_99_ values in diet-overlay bioassays was considered significantly different if their 95% fiducial limits did not overlap. The mortality percentage for each concentration was analyzed by one-way ANOVA, followed by post hoc Tukey HSD test. Statistical data analysis was based on the software package SPSS v.20 (IBM Corporation, Armonk, NY, USA). The percentage of larval growth inhibition was calculated using the formula below based on [40]:$$\mathrm{Growth}\;\mathrm{inhibition}\;(\%)=\frac{\;\mathrm{weight}\;\mathrm{of}\;\mathrm{the}\;\mathrm{control}\;\mathrm{larva}\;-\;\mathrm{weight}\;\mathrm{of}\;\mathrm{the}\;\mathrm{survivor}\;\mathrm{from}\;\mathrm{treatment}}{\mathrm{weight}\;\mathrm{of}\;\mathrm{the}\;\mathrm{control}\;\mathrm{larva}}\;\times \;100$$

## Figures and Tables

**Figure 1 toxins-13-00335-f001:**
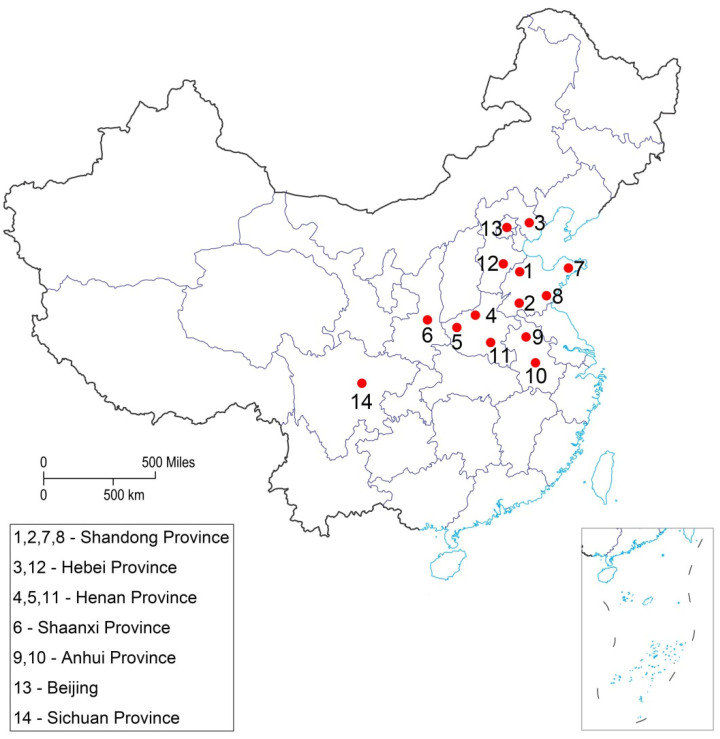
*Conogethes punctiferalis* sampling locations for baseline susceptibility study. The numbers on the map correspond to the location mentioned in Table 1.

**Table 1 toxins-13-00335-t001:** Source description of *Conogethes punctiferalis* populations used to establish baseline susceptibility to Bt Cry1Ab protein.

Province	Location	Coordinates	Collected Month	Number of Collected Larvae
Shandong	Dezhou (1)	37°26′10″ N; 116°21′32″ E	August 2019	250
Shandong	Jining (2)	35°24′54″ N; 116°35′14″ E	August 2019	350
Hebei	Tangshan (3)	39°37′52″ N; 118°10′48″ E	August 2019	350
Henan	Luoyang (4)	34°37′11″ N; 112°27′14″ E	September 2019	300
Henan	Jiyuan (5)	35°04′01″ N; 112°36′07″ E	September 2019	200
Shaanxi	Xi’an (6)	35°04′01″ N; 112°36′07″ E	September 2019	300
Shandong	Yantai (7)	37°28′35″ N; 121°26′26″ E	September 2020	340
Shandong	Linyi (8)	35°6′16.82″ N; 118°21′23.21″ E	September 2020	270
Anhui	Suzhou (9)	33°38′46.89″ N; 116°57′51.69″ E	September 2020	230
Anhui	Hefei (10)	31°51′50″ N; 117°16′50″ E	September 2020	220
Henan	Luohe (11)	33°34′53.09″ N; 114°0′59.54″ E	September 2020	320
Hebei	Handan (12)	36°37′32.37″ N; 114°32′20.26″ E	September 2020	330
Beijing	Shunyi (13)	40°7′49.25″ N; 116°39′16.74″ E	September 2020	340
Sichuan	Chengdu (14)	30°39′25.2000″ N; 104°3′57.6072″ E	October 2020	65

All YPM larvae across China were collected in the maize field.

**Table 2 toxins-13-00335-t002:** Diet-overlay bioassay to Bt Cry1Ab protein of fourteen field populations of *Conogethes punctiferalis* collected in 2019 and 2020 in China and laboratory susceptible population.

Year	Province	Location	g	*n*	Slope ± SE	LC_50_ (95% FL) ng/cm^2^ *	LC_99_ (95% FL) ng/cm^2^ *	*x* ^2^
2019	Shandong	Dezhou (1)	F1	576	0.98 ±.071	1.48 (1.05–2.10) bc	350.61 (160.66–967.20) b	7.80
	Shandong	Jining (2)	F1	576	1.32 ± 0.10	1.04 (0.78–1.40) bc	59.42 (32.67–129.88) ab	8.32
	Hebei	Tangshan (3)	F1	576	1.12 ± 0.12	2.38 (1.52–3.49) c	282.32 (125.59–951.33) b	8.38
	Henan	Luoyang (4)	F1	576	0.93 ± 0.07	0.55 (0.39–0.79) ab	178.28 (79.98–506.09) ab	8.96
	Henan	Jiyuan (5)	F2	576	1.03 ± 0.07	0.65 (0.47–0.91) ab	117.92 (56.96–302.38) ab	9.44
	Shaanxi	Xi’an (6)	F1	576	1.06 ± 0.08	0.88 (0.64–1.22) b	136.95 (67.55–340.90) ab	9.48
2020	Shandong	Yantai (7)	F1	576	1.16 ± 0.14	1.76 (1.05–2.64) bc	181.37 (80.55–654.10) ab	9.46
	Shandong	Linyi (8)	F1	576	1.02 ± 0.07	1.72 (1.23–2.42) c	321.31 (150.70–860.99) b	14.37
	Hebei	Handan (9)	F1	576	1.19 ± 0.12	2.19 (1.50–3.10) c	199.27 (96.07–563.99) ab	9.83
	Henan	Luohe (10)	F1	576	0.97 ± 0.07	0.35 (0.25–0.48) a	87.94 (40.98–237.27) ab	7.40
	Anhui	Suzhou (11)	F1	576	1.83 ± 0.28	2.14 (1.38–2.96) c	40.13 (21.97–114.96) a	6.06
	Anhui	Hefei (12)	F1	576	1.16 ± 0.09	1.60 (1.18–2.20) bc	163.83 (83.12–397.19) ab	11.74
	Beijing	Shunyi (13)	F1	576	0.93 ± 0.07	1.32 (0.93–1.88) bc	409.65 (182.33–1171.81) b	7.03
	Sichuan	Chengdu (14)	F3	576	1.20 ± 0.13	0.97 (0.61–1.42) bc	84.63 (41.05–247.33) ab	5.56
Population pooled from 2019 collection			F2	576	1.39 ± 0.11	0.92 (0.70–1.22) b	43.33 (24.28–92.73) a	9.50
Laboratory strain			F19	576	0.93 ± 0.07	1.08 (0.76–1.55) bc	349.77 (154.29–1012.84) b	10.46

g = Generation. *n* = Number of tested larvae. SE = Standard error. 95% FL = 95% fiducial limits. * = Values followed by the same lowercase letter in the same column indicate no significant difference (overlapping 95% fiducial limits). Degree of freedom (df) in each population is 19.

**Table 3 toxins-13-00335-t003:** *Conogethes punctiferalis* larval growth inhibition was observed for different Bt Cry1Ab concentrations after one week of treatment.

Cry1Ab (ng/cm^2^)		Larval Growth Inhibition (%)													Mean ± SEM
	Dezhou	Jining	Yantai	Linyi	Tang-shan	Handan	Luo-yang	Jiyuan	Luohe	Xi’an	Suzhou	Hefei	Shunyi	Cheng-du	
0.004	51.17	52.26	51.52	52.41	54.02	54.35	54.52	58.89	54.49	57.91	52.29	52.21	51.69	55.94	53.83 ± 0.64
0.02	57.00	61.45	57.86	58.47	63.44	61.97	61.13	62.19	61.60	59.85	55.92	54.47	57.90	59.86	59.51 ± 0.71
0.11	61.68	66.23	61.85	63.23	69.73	66.47	70.67	70.59	70.45	70.67	63.12	63.60	60.46	62.96	65.84 ± 1.04
0.55	76.10	70.00	76.96	75.70	74.11	73.79	74.59	72.62	74.90	71.23	75.61	75.66	79.10	77.42	74.84 ± 0.65
2.74	85.01	90.83	86.21	85.12	93.89	94.05	88.98	88.79	92.85	92.19	83.35	83.43	86.21	91.36	88.73 ± 1.03
13.68	99.78	99.98	99.79	99.76	99.84	99.92	99.91	99.67	99.91	99.85	99.78	99.80	99.70	99.83	99.82 ± 0.02
68.42	99.99	100.00	100.00	100.00	99.99	100.00	100.00	100.00	100.00	100.00	100.00	100.00	99.99	100.00	100.00 ± 0.00

The number of tested larvae in each location is 576. SEM = Standard error mean. Growth inhibition (%) = (Weight of the control larva—Weight of the survivors from treatment)/Weight of the control larva × 100 [40].

**Table 4 toxins-13-00335-t004:** *Conogethes punctiferalis* mortality % in the lower limit of LC_99_, LC_99_ and upper limit of LC_99_ of Bt Cry1Ab protein.

Location	*n*	Mortality %		
		24.28 ng/cm^2^ *	43.33 ng/cm^2^ *	92.73 ng/cm^2^ *
Yantai	240	80.83a	93.33a	99.17a
Linyi	240	81.67a	94.58a	99.58a
Handan	240	82.50a	95.00a	99.58a
Luohe	240	82.92a	95.42a	100.00a
Suzhou	240	81.25a	94.17a	99.17a
Hefei	240	81.25a	93.33a	100.00a
Shunyi	240	82.08a	95.42a	100.00a
Chengdu	240	81.67a	95.00a	100.00a

*n* = Number of larvae tested for each candidate concentration. * = Values followed by the same lowercase letter in the same column represent no significant difference at *p* > 0.05 (post hoc Tukey HSD test) by one-way ANOVA in SPSS. (Mortality % = (Number of dead larvae/Total number of tested larvae) × 100.

**Table 5 toxins-13-00335-t005:** Selection of resistance levels of *Conogethes punctiferalis* laboratory strain to Bt Cry1Ab protein at different generation.

Generation	Concentration ng/cm^2^	LC_50_ (95% FL) ng/cm^2^	LC_90_ (95% FL) ng/cm^2^	Slope ± SE	RR	*x* ^2^
Cry1AbS		1.08 (0.76–1.55)	26.07 (15.17–51.59)	0. 93 ± 0.07		10.46
F1	1.08	1.14 (0.82–1.59)	19.58 (12.07–35.89)	1.04 ± 0.08	1.06	7.29
F2	1.08	1.78 (1.25–2.58)	45.67 (25.99–93.42)	0.91 ± 0.07	1.65	4.50
F3	1.08	2.44 (1.61–3.55)	36.51 (22.04–72.45)	1.09 ± 0.11	2.26	9.90
F4	1.08	3.07 (2.233–4.08)	19.53 (13.42–32.83)	1.59 ± 0.17	2.84	7.96
F5	1.08	3.30 (2.37–4.66)	56.36 (33.70–108.74)	1.04 ± 0.08	3.06	9.76
F6	2.05	3.57 (2.58–4.87)	32.64 (21.34–57.60)	1.33 ± 0.13	3.30	6.97
F7	2.05	4.11 (2.94–5.68)	44.10 (27.95–81.19)	1.24 ± 0.12	3.81	6.51
F8	2.05	5.03 (3.42–7.35)	87.71 (49.78–191.88)	1.03 ± 0.10	4.67	7.05
F9	3.12	5.14 (3.57–7.61)	121.85 (66.25–269.96)	0.93 ± 0.07	4.76	7.62
F10	3.12	5.42 (3.80–7.97)	112.62 (62.34–243.97)	0.97 ± 0.08	5.02	9.01
F11	4.32	5.85 (3.94–8.59)	100.71 (56.21–230.14)	1.04 ± 0.11	5.42	5.91
F12	4.32	6.02 (4.17–9.01)	147.30 (78.36–338.77)	0.92 ± 0.08	5.57	5.94
F13	6.16	6.56 (4.04–11.59)	581.99 (224.53–2131.07)	0.66 ± 0.06	6.08	13.19
F14	6.16	7.67 (5.22–11.78)	210.74 (106.67–523.77)	0.89 ± 0.08	7.10	5.02

g = Generation. Cry1AbS = Cry1Ab susceptible strain. *n* = Number of tested larvae. df = Degree of freedom. RR = Resistant ratio is the LC_50_ value of resistance strain divided by the LC_50_ value of susceptible strain. Degree of freedom (df) is 19 and the number of tested larvae in each generation is 576.

## Data Availability

Not applicable.

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
