# Peer review of "Baseline Susceptibility and Laboratory Selection of Resistance to Bt Cry1Ab Protein of Chinese Populations of Yellow Peach Moth, Conogethes punctiferalis (Guenée)"

_toxins, 2021, doi:10.3390/toxins13050335_

Round 1

Reviewer 1 Report

Dear Authors, dear Editor,

draft “Baseline Susceptibility of Yellow Peach Moth, … to Bt Cry1Ab, toxins-1183827” describes a mapping study performed in some regions of the PRC on the resistance of the YPM to the Bacillum thuringensis toxin. This work is motivated since a transgenic maize that expresses the toxin as an embedded pesticide will be marketed in the country and it is necessary to know in advance, whether and how much resistance to this biological pesticide has already developed and will develop in the local moth strains. The researchers sampled the moths from the different places and performed on them the tests.

In Figure 1, what does the small map rectangle without sampling positions indicate? If you eliminate that, you can show a larger map (use larger numbers).

It would be interesting to readers to understand whether Bt-maize (or any other GM Bt crop) has ever been used in the region, because it is unusual that resistance develops before pesticide use.

In Table 2, the columns headed “g” and “df” have always the same value and should be eliminated, with the datum reported in the notes. Example: “163.83(83.12- 397.19)ab ” where are notes a and b? in adition, given tne order of magniturde of values, in LC99 there are too many decimal places: keep to the integer.

In Table 3, as well, two digits for the percent is too much: keep to the integer. The same problem is also in the text. In fact, in Table 4, differences are NS although % is reported with two decimal digits (one part in 10.000 is too optimistic).

In Table 5, the columns headed “n” and “df” have always the same value and should be eliminated, with the datum reported in the notes.

I would advise to edit the text for clarity and to present results in a more compact way.

With such revision, the draft may be published.

Best regards

Author Response

Reviewers 1 suggestions and comments are addressed in the attached file

Reviewer 2 Report

This manuscript reported the establishment of baseline susceptibility to Bt Cry1Ab protein of a major maize pest Yellow Peach Moth (YPM), Conogethes punctiferalis. Different geographical populations of YPM from the main maize-growing regions in China were sampled and tested, and a pooled laboratory population was used for multi-generational resistance study. The study was well carried out and the results obtained here serve a critical role for the application of Bt maize for the control of this pest. However, there are some concerns or flaws in this manuscript and should be addressed before consideration for publication.

Introduction: it is not elaborated enough how YPM became the most destructive maize pest in China in recent years, what are the history and current status of the pest control practice for YPM in those regions, why Bt-maize is needed and how this can be incorporated into an integrative pest management of this pest. Details for these points are not needed, but efforts should be made to clarify them so people could better understand the background and results of this study. Results: some tables can be improved by adjusting the column width for better presentation, and some more data from the tables should be described or compared (see comments below). Discussion: population genetics and seasonal migration of YPM should be taken into consideration to interpret the results here. In addition, the writing of this manuscript is far from optimal. There are quite some excessive uses of words, unclear descriptions of context and incorrect English, which should be improved. Please see more detailed comments below:

Line 2-4: the title is misleading and a bit too long. Consider to use “Baseline Susceptibility and Laboratory Selection of Resistance to Bt Cry1Ab Protein of Chinese Populations of  Yellow Peach Moth, Conogethes punctiferalis (Guenée)”.

Line 10-11: change this sentence into “To use Bt maize against YPM, the baseline susceptibility of the local populations in the targeted areas needs to be verified”, which should point out the target species and link to the next sentence. 

Line 19 and 28: consider to use “development” instead of “evolution” here and some other places in the manuscript. Conclusion for “evolution” can hardly be made here considering the time scale of the experiments.

Line 33-34: add “as” before “an alternative”, and add “the orders” before “Lepidoptera”.

Line 42: change “consequences” into “concerns”, change “the possibility of target insects evolving” into “possible insect”,

Line 44: it’s rather rare to use 16 citations (12-27) to support one sentence. So here better to select some references that are either reviews or classic.

Line 45-46: change “using new technology” into “growing of Bt crops”, and change “transgenic maize” into “product”.

Line 47-48: it’s not correct English here and should be rewritten.

Line 48-51: this is also obscure and should be rewritten.

Line 54: “result in” or “result from”?

Line 62-69: is this a general statement (worldwide) or specifically describe the situation in the Huang-Huai-Hai Summer (HHHS) Maize Region? The authors should clarify this and if it is a general statement, it’s would be better if giving more specific information about HHHS region such as pest composition, planting history and current pest control practices.

Line 72-74: people may be wondering why YPM has become the dominate pest in HHHS region among all Lepidopteran pests but not before. Thus, it would be helpful if the authors can give a brief explanation or speculation.

Line 74-75: move “and” before “causing”.

Line 76-77: better to use “crucial” instead of “vital”, change “target and against” into “control”.

Line 77: “well-supported” is cryptic, I assume the authors meant “science-based”?

Line 79-82: change “for YPM populations from geographically different locations across China's major maize growing regions and” into “of different YPM geographical populations across major maize growing regions in China, which”.

Line 88: change “because of” into “which is the”.

Line 92: add “for the populations” before “over the two years”.

Figure 1 and table 1: either figure 1 or table 1 can be made into supplementary file since they are overlapping with each other. Table 1 fits more into supplementary for method (sample collection). The number corresponds to the location in Fig.1 should be shown in table 1 (either in the brackets after the province, or in an additional column). The life stage and host can be removed from table 1 and put into table notes, since there are no variables for them.

Line 94-98: More comparison/results need to be described and discussed accordingly. E.g., why the LC50s were in general (significantly) lower from the inner land (Henan, Shaanxi or Sichuan) than from east regions (Shandong, Hebei or Anhui); why LC99 from Suzhou was significantly higher than others; why LC99 of F19 was significantly higher than that from F2 in the pooled population.

Table 2: add the statistics method into the table note.

Table 3: the space among columns should be increased for better presentation (currently the city names are almost connected which is confusing). Why statistics was not used here?

Table 4: the authors used lower case letter for statistics in the previous table but here use “ns”, but it’s no clear (at least from the table) if the data were compared only for location or treatment or both. Again, the statistics method (one way or two-way ANOVA, or something else) should be noted here.

Table 5: what is “S” in the first column? Susceptible or control? Then it should be explained in the table note. Why there are two rows of data for RR in each generation? Or the number below is the last digit of the number above? Then it should be checked and avoided! Delete a space before 81.19 in F7, and add a space before 523.77 in F14.

Line 144-145: change “widespread commercialization” into “broad commercial deployment”, and change “initiation” into “establishment”.

Line 147: change “establish” into “verify”.

Line151-153:  since here the variations in LC50 and LC99 were mentioned, it should be followed by the paragraph below (160-178) as the first paragraph in discussion to specific address the variations in baseline susceptibility.

Line 153-159: this point should be explored as the second paragraph in discussion. It is not necessary to show the actual data from the reference (line 155-158), instead simply cite their conclusion. However, it will be helpful to indicate the provinces for the 11 geographic populations from Zhang et al. (2010) to discuss the relation between the susceptibility to Cry1Ab and genetic diversity in certain geographic populations. For example, Zhang et al. (2010) concluded that high genetic similarity among 11 populations but Shandong populations showed lower genetic diversity than others and biggest genetic distance was identified between Sichuan and Shandong populations.  Here the LC50 of Shandong populations were in general higher than those from inner land provinces (although not significant). The authors can discuss if difference in genetic diversity would contribute to the difference in susceptibility.

Line 160: add “in baseline susceptibility” after “The natural variation”.

Line 164: delete “variations”.

Line 166: change “was found in” into “of”.

Line 179: change “consider” into “be considered”.

Line 181: change “to” into “by”, and change “produced” into “detected”.

Line 182-183: “reduced” in what way? The authors should discuss the difference in the data among concentrations and/or populations.

Line 186-187: “agreeing with our finding” in what way? The 0.82 ng/cm2 (gave >90 % inhibition) was not used in this study, while higher 2.74 ng/cm2 gave lower mean 88.7 % inhibition here. So strictly speaking they are not agreeing with each other. The authors should elaborate such claim.

Line 200-202: this sentence is not clear. What do you mean by “the development of…..be successfully managed”? do you mean the diagnostic concentration can be identified in the same way, or the same diagnostic concentration can be used for the two species? It should be rewritten to make the point clear.

Line 215-216: this sentence is not clear and should be separated into two sentences, to make it clear (briefly) why you want to continue the selection experiment and how you what to study the resistance mechanism. Actually, I would suggest delete this since it is not necessary.

Line 219: change “establishing” into “establish”, change “in maize” into “for YPM”.

Line 221-223: change “designated“ into “verified”, and delete the rest of the sentence after “diagnostic concentration”. In addition, the conclusion for other results (low variations, growth inhibition, laboratory selection) should also be summarized here.

Line 223-225: add “in the laboratory” after “being developed”, which is important. Change “and when available, they will be valid for testing this diagnostic technique's capacity to discriminate between susceptible and resistant populations.” into “and can be potentially used for testing the efficacy of the diagnostic technique”.

Line 225-227: I don’t know what this means and should be rewritten to make the point clear.

Line 230-231: delete “larval”, add “including” before “Shandong”, and delete “,” after “Anhui”.

Line 249: change “on“ into “using”.

Line 278-279: here it says “reared on the artificial diet in the laboratory for eighteen generations without exposure to Bt proteins”, then afterwards it mentioned “The population was initially treated with LC50 value of Cry1Ab”. The way I understand it is that there was a susceptible strain that kept in parallel. The authors should make it clear which strain was used for which experiments. E.g., the susceptible strain should be mentioned in the notes for table 2, but not here in “Laboratory selection”.

Line 292-293: it didn’t mention the statistics for table 4.

Author Response

Reviewer 2 suggestions and comments are addressed in the attached file.
